# Tree-Structured Attention with Hierarchical Accumulation

**Xuan-Phi Nguyen**[‡][*], **Shafiq Joty**[†][‡], **Steven C.H. Hoi**[†], **Richard Socher**[†]
[†]Salesforce Research
[‡]Nanyang Technological University
nguyenxu002@e.ntu.edu.sg,{sjoty,shoi,rsocher}@salesforce.com

## Abstract

Incorporating hierarchical structures like constituency trees has been shown to be effective for various natural language processing (NLP) tasks. However, it is evident that state-of-the-art (SOTA) sequence-based models like the Transformer struggle to encode such structures inherently. On the other hand, dedicated models like the Tree-LSTM, while explicitly modeling hierarchical structures, do not perform as efficiently as the Transformer. In this paper, we attempt to bridge this gap with "Hierarchical Accumulation" to encode parse tree structures into self-attention at constant time complexity. Our approach outperforms SOTA methods in four IWSLT translation tasks and the WMT'14 English-German translation task. It also yields improvements over Transformer and Tree-LSTM on three text classification tasks. We further demonstrate that using hierarchical priors can compensate for data shortage, and that our model prefers phrase-level attentions over token-level attentions.

## 1 Introduction

Although natural language has a linear surface form, the underlying construction process is known to be hierarchical (Frege, 1892). As such, different tree-like structures have been proposed to represent the compositional grammar and meaning of a text, such as constituency and dependency trees. Leveraging the hierarchical structures of language gives models more structural information about the data and improves performance on the downstream tasks (Tai et al., 2015; Eriguchi et al., 2016). Despite that, state-of-the-art neural models like the Transformer still prefer the linear (sequential) form of natural language (Vaswani et al., 2017; Ott et al., 2018; Devlin et al., 2018). This is because the linear form allows us to develop simple but efficient and scalable techniques (like *self-attention* which operates at *constant parallel time complexity*[1]) to train models at a large scale. Yet, there is still no concrete evidence that these models learn grammatical and constituency structures implicitly. However, ad hoc tree-structured models (Socher et al., 2013; Tai et al., 2015; Shi et al., 2018) often operate on recursive or recurrent mechanisms, which are not parallelizable, thus hindering their application in larger-scale training. Besides, such models are designed to only operate at the sentence-level (i.e., single tree), limiting their application to document-level processing.

We propose a novel attention-based method that encodes trees in a bottom-up manner and executes competitively with the Transformer at constant parallel time complexity. In particular, our attention layers receive as input the constituency tree of a piece of text and then model the hidden states of all nodes in the tree (leaves and nonterminals) from their lower-layer representations according to the tree structure. As attentions typically have query, key and value components, our model uses *hierarchical accumulation* to encode the *value* component of each nonterminal node by aggregating the hidden states of all of its descendants. The accumulation process is three-staged. First, we induce the value states of nonterminals with *hierarchical embeddings*, which help the model become aware of the hierarchical and sibling relationships between the nodes. Second, we perform an *upward cumulative-average* operation on each target node, which accumulates all elements in the branches originating from the target node to its descendant leaves. Third, these branch-level representations

---

[*]Work done during an internship at Salesforce Research Asia, Singapore.

[1]Given GPUs, a "constant parallel time" process can perform all its computations at once algorithmically.

are combined into a new value representation of the target node by using *weighted aggregation*. Finally, the model proceeds to perform attention with *subtree masking* where the attention score between a nonterminal query and a key is activated only if the key is a descendant of the query.

Our contributions are threefold. First, we present our attention-based hierarchical encoding method. Our method overcomes linear parallel time complexity of Tree-LSTM (Tai et al., 2015) and offers attractive scalability. Second, we adopt our methods within the Transformer architecture and show improvements across various NLP tasks over strong baselines. In particular, our model leverages tree-based prior to improve translation quality over the Transformer baselines in the IWSLT'14 English-German and German-English, the IWSLT'13 English-French and French-English, and the WMT'14 English-German translation tasks. Furthermore, our model also exhibits advantages over Tree-LSTM in classification tasks including Stanford Sentiment Analysis (SST) (Socher et al., 2013), IMDB Sentiment Analysis and Subject-Verb Agreement (Linzen et al., 2016). Finally, our analysis of the results suggests that incorporating a hierarchical prior using our method can compensate for the lack of data in the context of machine translation. We also demonstrate that the model has natural and consistent preference for phrase-level attention over token-level attention. Our source code is available at https://github.com/nxphi47/tree_transformer.

## 2 RELATED WORK

The Transformer framework has become the driving force in recent NLP research. For example, it has achieved state-of-the-art performance in machine translation tasks (Vaswani et al., 2017; Shaw et al., 2018; Ott et al., 2018; Wu et al., 2019) and self-supervised representational learning (Devlin et al., 2018; Radford et al., 2018; Lample & Conneau, 2019; Yang et al., 2019). The self-attention layers in the Transformer encode a *sequence* at *constant* parallel time complexity, which makes it parallelizeable and scalable. On the other hand, there have been many proposals to use *parse trees* as an architectural prior to facilitate different downstream tasks. Socher et al. (2013) adopt a recursive compositional method over constituency trees to solve sentiment analysis in a bottom-up manner. Tree-LSTM (Tai et al., 2015) improves the task performance by using an LSTM structure to encode trees recurrently. Both of the proposed methods, while effective, operate *sequentially* in parallel time complexity. Tree structures have also been used as an architectural bias to improve machine translation (Eriguchi et al., 2016; Shi et al., 2018; Yang et al., 2017). Constituency trees can also be decoded in a top-down manner, as proposed in (Alvarez-Melis & Jaakkola, 2017; Gū et al., 2018). Besides, they can also be learned in an unsupervised way (Kim et al., 2019; Shen et al., 2018; 2019; Yaushian Wang & Chen, 2019). Meanwhile, Strubell et al. (2018); Hao et al. (2019); Harer et al. (2019) attempted to incorporate trees into self-attention. Hewitt & Manning (2019) showed that *dependency* semantics are already intrinsically embedded in BERT (Devlin et al., 2018). Concurrently, Yaushian Wang & Chen (2019) suggested that BERT may not naturally embed *constituency* semantics.

Our approach encodes trees in a bottom-up manner as Tai et al. (2015) and Socher et al. (2013). But it differs from them in that it leverages the attention mechanism to achieve high efficiency and performance. Plus, it is applicable to self- and cross-attention layers in the Transformer sequence-to-sequence (Seq2Seq) skeleton. Unlike previous methods, our model works with multi-sentence documents (multi-tree) seamlessly. Our model also differs from Strubell et al. (2018); Hao et al. (2019); Harer et al. (2019) in that their methods only use tree structures to guide and mask token-level attentions, while ours processes all the nodes of the tree hierarchically. In this paper, while applicable to dependency trees, our approach focused primarily on *constituency* trees because (1) constituency trees contain richer grammatical information in terms of phrase structure, and (2) there is as yet no evidence, to the best of our knowledge, that constituency structures are learned implicitly in the standard self-supervised models.

## 3 BACKGROUND - TRANSFORMER FRAMEWORK

The Transformer (Vaswani et al., 2017) is a Seq2Seq network that models sequential information using stacked self- and cross-attention layers. The output $O$ of each attention sub-layer is computed via scaled multiplicative formulations defined as:

$$A = (QW^Q)(KW^K)^T/\sqrt{d}; \qquad \text{Att}(Q, K, V) = \text{softmax}(A)(VW^V) \qquad (1)$$

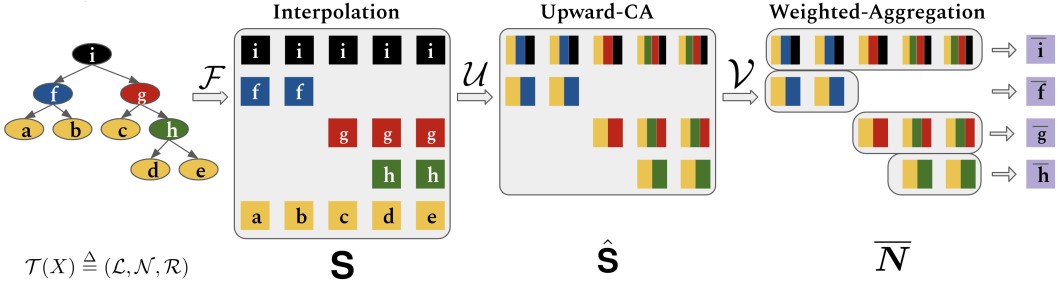

Figure 1: The hierarchical accumulation process of tree structures (best seen in colors). Given a parse tree, it is interpolated into a tensor **S**, which is then accumulated vertically from bottom to top to produce **Ŝ**. Next, the (branch-level) component representations of the nonterminal nodes are combined into one representation as $\overline{N}$ by weighted aggregation. Multi-colored blocks indicate accumulation of nodes of respective colors. The rows of **S** in Eq. 5 are counted from the bottom.

$$O = \text{Att}(Q, K, V)W^O \qquad (2)$$

where softmax is the *softmax* function, $Q = (q_1, ..., q_{l_q}) \in \mathbb{R}^{l_q \times d}$, $K = (k_1, ..., k_{l_k}) \in \mathbb{R}^{l_k \times d}$, $V = (v_1, ..., v_{l_k}) \in \mathbb{R}^{l_k \times d}$ are matrices of query, key and value vectors respectively, and $W^Q, W^K, W^V, W^O \in \mathbb{R}^{d \times d}$ are the associated trainable weight matrices. $A$ denotes the *affinity* scores (attention scores) between queries and keys, while $\text{Att}(Q, K, V)$ are the attention vectors. Then, the final output of a Transformer layer is computed as:

$$\phi(A, Q) = \text{LN}(\text{FFN}(\text{LN}(O + Q)) + \text{LN}(O + Q)) \qquad (3)$$

where $\phi$ represents the typical serial computations of a Transformer layer with layer normalization (LN) and feed-forward (FFN) layers. For simplicity, we omit the multi-head structure and other details and refer the reader to Vaswani et al. (2017) for a complete description.

## 4 TREE-BASED ATTENTION

### 4.1 ENCODING TREES WITH HIERARCHICAL ACCUMULATIONS

To encode hierarchical structures in parallel, we need to represent the tree in a data structure that can be parallellized. Given a sentence $X$ of length $n$, let $\mathcal{G}(X)$ be the directed spanning tree which represents the parse tree of $X$ produced by a parser. We define a transformation $\mathcal{H}$ such that $\mathcal{H}(\mathcal{G}(X)) = \mathcal{T}(X) \triangleq (\mathcal{L}, \mathcal{N}, \mathcal{R})$. In this formulation, $\mathcal{L}$ denotes the *ordered sequence* of $n$ terminal nodes (or *leaves*) of the tree (i.e., $\mathcal{L} = X$), and $\mathcal{N}$ denotes the set of $m$ nonterminal nodes (or simply *nodes*), each of which has a phrase label (e.g., NP, VP) and spans over a sequence of terminal nodes.[2] $\mathcal{R}$ contains a set of *rules* indexed by the nonterminal nodes in $\mathcal{N}$ such that for each node $x \in \mathcal{N}$, $\mathcal{R}(x)$ denotes the set of all nodes that belong to the subtree rooted at $x$. For example, for the nonterminals $g$ and $h$ in Figure 1, $\mathcal{R}(g) = \{g, c, h, d, e\}$ and $\mathcal{R}(h) = \{h, d, e\}$.

There might be various ways to transform the tree $\mathcal{G}(X)$. For a tree-encoding process, a particular transformation is legitimate only if the resulting data structure represents only $\mathcal{G}(X)$ and not any other structures. Otherwise, the encoding process may confuse $\mathcal{G}(X)$ with another structure. In other words, the transformation should be a one-to-one mapping. Our transformation $\mathcal{H}$ satisfies this requirement as shown in the following proposition (see Appendix 7.1 for a proof).

**Proposition 1** *Suppose $\mathcal{G}(X)$ is a parse tree and there exists a inverse-transformation $\mathcal{I}$ that converts $\mathcal{T}(X)$ to a graph $\mathcal{I}(\mathcal{T}(X))$, then $\mathcal{I}$ can only transform $\mathcal{T}(X)$ back to $\mathcal{G}(X)$, or:*

$$\mathcal{I}(\mathcal{H}(\mathcal{G}(X))) = \mathcal{G}(X) \quad \text{or} \quad \mathcal{I} = \mathcal{H}^{-1} \qquad (4)$$

We now describe the tree accumulation method using $\mathcal{T}(X)$. Figure 1 shows the overall process. Let $L = (l_1, ..., l_n) \in \mathbb{R}^{n \times d}$ and $N = (n_1, ..., n_m) \in \mathbb{R}^{m \times d}$ be the hidden representations of the

---

[2]We omit the part-of-speech tags of the words which constitute the preterminal nodes in a constituency tree.

leaves $\mathcal{L} = (x_1^{\mathcal{L}}, ..., x_n^{\mathcal{L}})$ and nodes $\mathcal{N} = (x_1^{\mathcal{N}}, ..., x_m^{\mathcal{N}})$, respectively. We define an *interpolation function* $\mathcal{F} : (\mathbb{R}^{n \times d}, \mathbb{R}^{m \times d}) \rightarrow \mathbb{R}^{(m+1) \times n \times d}$, which takes $\boldsymbol{L}$, $\boldsymbol{N}$ and $\mathcal{R}$ as inputs and returns a tensor $\mathbf{S} \in \mathbb{R}^{(m+1) \times n \times d}$. The row $i$ and column $j$ vector of $\mathbf{S}$, or $\mathbf{S}_{i,j} \in \mathbb{R}^d$, is defined as:

$$\mathbf{S}_{i,j} = \mathcal{F}(\boldsymbol{L}, \boldsymbol{N}, \mathcal{R})_{i,j} = \begin{cases} \boldsymbol{l}_j & \text{if } i = 1 \\ \boldsymbol{n}_{i-1} & \text{else if } x_j^{\mathcal{L}} \in \mathcal{R}(x_{i-1}^{\mathcal{N}}) \\ \boldsymbol{0} & \text{otherwise.} \end{cases} \tag{5}$$

where $\boldsymbol{0}$ denotes a *zero* vector of length $d$. Note that the row and column arrangements in $\mathbf{S}$ reflect the tree structure (see Figure 1). Next, we perform the *upward cumulative-average (upward-CA)* operation $\mathcal{U}$ on $\mathbf{S}$ to compose the node representations in a bottom-up fashion over the induced tree structure. The result of this operation is a tensor $\hat{\mathbf{S}} \in \mathbb{R}^{m \times n \times d}$, in which each nonterminal node representation is averaged along with all of its descendants in a particular branch. More formally,

$$\mathcal{U}(\mathbf{S})_{i,j} = \hat{\mathbf{S}}_{i,j} = \begin{cases} \boldsymbol{0} & \text{if } \mathbf{S}_{i+1,j} = \boldsymbol{0} \\ \sum_{\mathbf{S}_{t,j} \in C_j^i} \mathbf{S}_{t,j} / |C_j^i| & \text{otherwise.} \end{cases} \tag{6}$$

where $C_j^i = \{\mathbf{S}_{1,j}\} \cup \{\mathbf{S}_{t,j} | x_t^{\mathcal{N}} \in \mathcal{R}(x_i^{\mathcal{N}})\}$ is the set of vectors in $\mathbf{S}$ representing the leaves and nodes in the branch that starts with $x_i^{\mathcal{N}}$ and ends with $x_j^{\mathcal{L}}$. Note that we discard the leaves in $\hat{\mathbf{S}}$. As demonstrated in Figure 1, each row $i$ of $\hat{\mathbf{S}}$ represents a nonterminal node $x_i^{\mathcal{N}}$ and each entry $\hat{\mathbf{S}}_{i,j}$ represents its vector representation reflecting the tree branch from $x_i^{\mathcal{N}}$ to a leaf $x_j^{\mathcal{L}}$. This gives $|\mathcal{R}(x_i^{\mathcal{N}}) \cap \mathcal{L}|$ different constituents of $x_i^{\mathcal{N}}$ that represent the branches rooted at $x_i^{\mathcal{N}}$.

The next task is to combine the branch-level accumulated representations of a nonterminal $x_i^{\mathcal{N}}$ into a single vector $\overline{\boldsymbol{n}}_i$ that encapsulates all the elements in the subtree rooted by $x_i^{\mathcal{N}}$. Our method does so with a *weighted aggregation* operation. The aggregation function $\mathcal{V}$ takes $\hat{\mathbf{S}}$ as input and a weighting vector $\boldsymbol{w} \in \mathbb{R}^n$, and computes the final node representations $\overline{\boldsymbol{N}} = (\overline{\boldsymbol{n}}_1, ..., \overline{\boldsymbol{n}}_m) \in \mathbb{R}^{m \times d}$, where each row-vector $\overline{\boldsymbol{n}}_i$ in $\overline{\boldsymbol{N}}$ is computed as:

$$\mathcal{V}(\hat{\mathbf{S}}, \boldsymbol{w})_i = \overline{\boldsymbol{n}}_i = \frac{1}{|\mathcal{L} \cap \mathcal{R}(x_i^{\mathcal{N}})|} \sum_{j : x_j^{\mathcal{L}} \in \mathcal{R}(x_i^{\mathcal{N}})} w_j \odot \hat{\mathbf{S}}_{i,j} \tag{7}$$

where $\odot$ denotes the element-wise multiplication. Specifically, the aggregation function $\mathcal{V}$ computes a weighted average of the branch-level representations. In summary, the hierarchical accumulation process can be expressed as the following equation:

$$\overline{\boldsymbol{N}} = \mathcal{V}(\mathcal{U}(\mathbf{S}), \boldsymbol{w}) = \mathcal{V}(\mathcal{U}(\mathcal{F}(\boldsymbol{L}, \boldsymbol{N}, \mathcal{R})), \boldsymbol{w}) \tag{8}$$

## 4.2 HIERARCHICAL EMBEDDINGS

While the above technique is able to model the states of nonterminal nodes as an encapsulation of their respective descendants, those descendants are equally treated since no biases are imposed on them. In other words, although each branch from a node comprises a distinctive set of descendants, the hierarchy of elements within a branch and the sibling relationship among branches are not explicitly represented. Thus, it may be beneficial to introduce biases that reflect such underlying subtree-level hierarchical structures. We propose *Hierarchical Embeddings* to induce distinguishable tree structures into the tensor $\mathbf{S}$ before being accumulated by $\mathcal{U}$ and $\mathcal{V}$. We also demonstrate the effectiveness of these embeddings with experiments in Section 5. Figure 2 illustrates the hierarchical embeddings for the nodes in Figure 1. Given $\boldsymbol{L}$, $\boldsymbol{N}$ and $\mathcal{R}$ as defined in Section 4.1, we construct a

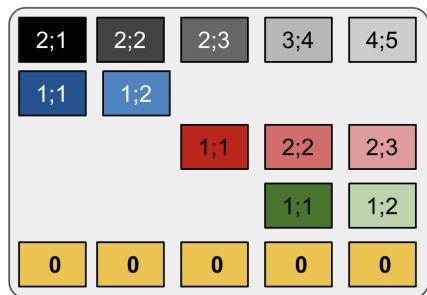

Figure 2: Hierarchical Embeddings. Each block $\mathbf{E}_{i,j}$ is an embedding vector $[e_x^v; e_y^h]$ with indices $x, y$ following the syntax "$x; y$", where $x = |V_j^i|$ and $y = |H_j^i|$. "$\boldsymbol{0}$" indicates no embedding.

tensor of hierarchical embeddings $\mathbf{E} \in \mathbb{R}^{(m+1)\times n \times d}$ with entries defined as follows:

$$\mathbf{E}_{i,j} = \begin{cases} [e^v_{|V^i_j|}; e^h_{|H^i_j|}] & \text{if } i > 1 \text{ and } x^{\mathcal{L}}_j \in \mathcal{R}(x^{\mathcal{N}}_i) \\ \mathbf{0} & \text{otherwise.} \end{cases} \tag{9}$$

where $V^i_j = \{x^{\mathcal{N}}_t | x^{\mathcal{N}}_t \in \mathcal{R}(x^{\mathcal{N}}_i) \text{ and } x^{\mathcal{L}}_j \in \mathcal{R}(x^{\mathcal{N}}_t)\}$ is the set of $x^{\mathcal{L}}_j$'s ancestors up to $x^{\mathcal{N}}_i$, and $H^i_j = \{x^{\mathcal{L}}_t | t \le j \text{ and } x^{\mathcal{L}}_t \in \mathcal{L} \cap \mathcal{R}(x^{\mathcal{N}}_i)\}$ is the set of leaves from the leftmost leaf up to $x^{\mathcal{L}}_j$ of the $x^{\mathcal{N}}_i$-rooted subtree; $e^v_i$ and $e^h_i$ are embedding row-vectors of the respective trainable *vertical* and *horizontal* embedding matrices $\boldsymbol{E}^v, \boldsymbol{E}^h \in \mathbb{R}^{|E| \times \frac{d}{2}}$ and $[\bullet; \bullet]$ denotes the concatenation operation in the hidden dimension. The vertical embeddings represent the path length of a node to a leaf which expresses the hierarchical order within a branch, whereas the horizontal embeddings exhibit the relationship among branch siblings in a subtree. The resulting node representations after hierarchical encoding are defined as:

$$\overline{\boldsymbol{N}}' = \mathcal{V}(\mathcal{U}(\mathbf{S}+\mathbf{E}), \boldsymbol{w}) \tag{10}$$

Note that we share such embeddings across attention heads, making them account for only $0.1\%$ of the total parameters (see Appendix 7.3 for more information).

### 4.3 SUBTREE MASKING

Masking attentions is a common practice to filter out irrelevant signals. For example, in the decoder self-attention layers of the Transformer, the affinity values between query $\boldsymbol{q}_i$ and key $\boldsymbol{k}_j$ are turned off for $j > i$ to avoid future keys being attended since they are not available during inference. This can be done by adding to the affinity $\boldsymbol{q}^T_i \boldsymbol{k}_j$ an infinitely *negative* value ($-\infty$) so that the resulting attention weight (after *softmax*) becomes *zero*.

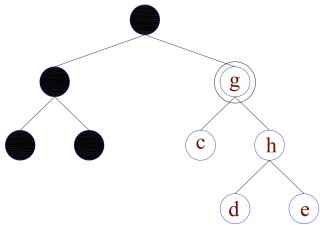

Figure 3: Subtree masking. Given the query at position $g$, attentions are only included within the $g$-rooted subtree, while the remaining elements are masked out (shaded).

In the context of tree-based attentions (to be described next), we promote the bottom-up structure by introducing *subtree masking* for encoder self-attention.[3] That is, if a node-query $q^{\mathcal{N}}_i \in \mathcal{N}$ is attending to a set of node-keys $k^{\mathcal{N}}_j \in \mathcal{N}$ and leaf-keys $k^{\mathcal{L}}_j \in \mathcal{L}$, attentions are turned **on** only for affinity pairs whose key belongs to the *subtree* rooted at $q^{\mathcal{N}}_i$. In other words, each node-query has access only to its own subtree descendants, but not to its ancestors and siblings. On the other hand, if a leaf-query $q^{\mathcal{L}}_i \in \mathcal{L}$ is attending, only leaf-keys are turned on, like in the Transformer. Figure 3 illustrates the subtree masking with an example. More formally, given $a_{ij}$ as the affinity value between a node/leaf-query $q_i \in \mathcal{N} \cup \mathcal{L}$ and a node/leaf-key $k_j \in \mathcal{N} \cup \mathcal{L}$, the masking function $\mu$ is defined as:

$$\mu(a_{ij}) = \begin{cases} a_{ij} & \text{if } (q_i \in \mathcal{N} \text{ and } k_j \in \mathcal{R}(q_i)) \text{ or } (q_i, k_j \in \mathcal{L}) \\ a_{ij} - \infty & \text{otherwise.} \end{cases} \tag{11}$$

### 4.4 INTEGRATING INTO TRANSFORMER FRAMEWORK

In this section, we describe how the above proposed methods fit into self- and cross-attentions of the Transformer framework, which enable them to efficiently encode parse trees.

**Encoder Self-attention.** Figure 4a visualizes the encoder self-attention process. Without loss of generality, let $\boldsymbol{L} \in \mathbb{R}^{n \times d}$ and $\boldsymbol{N} \in \mathbb{R}^{m \times d}$ respectively denote the leaf and node representations that a Transformer encoder layer receives from its previous layer along with the parse tree represented as $\mathcal{T}(X) = (\mathcal{L}, \mathcal{N}, \mathcal{R})$.[4] The tree-based self-attention layer then computes the respective output representations $\hat{\boldsymbol{L}}$ and $\hat{\boldsymbol{N}}$. Specifically, first, we *compare* the node and leaf representations against

---

[3]It can only be done in the encoder self-attention.

[4]For the first layer, $\boldsymbol{L}$ and $\boldsymbol{N}$ are token embeddings; $\boldsymbol{L}$ also has positional encodings.

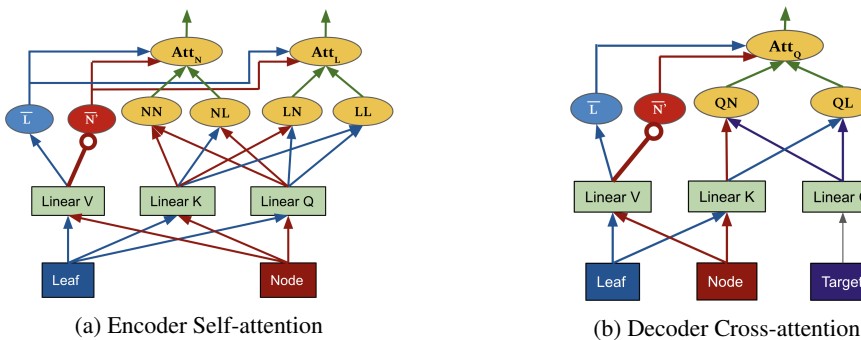

(a) Encoder Self-attention        (b) Decoder Cross-attention

Figure 4: Illustration of the proposed Tree-based Attentions: (a) Encoder self-attention, (b) Decoder cross-attention. Circle-ended arrows indicate where hierarchical accumulations take place. The overall Transformer architecture is provided in Figure 6 (Appendix 7.3).

each other to produce query-key affinity matrices $\boldsymbol{A}_{NL} \in \mathbb{R}^{m \times n}$, $\boldsymbol{A}_{NN} \in \mathbb{R}^{m \times m}$, $\boldsymbol{A}_{LL} \in \mathbb{R}^{n \times n}$ and $\boldsymbol{A}_{LN} \in \mathbb{R}^{n \times m}$ for node-leaf (i.e., node representation as the query and leaf representation as the key), node-node, leaf-leaf, and leaf-node pairs, respectively, as follows:

$$\boldsymbol{A}_{NL} = (\boldsymbol{N}\boldsymbol{W}^Q)(\boldsymbol{L}\boldsymbol{W}^K)^T/\sqrt{d} \qquad (12) \qquad \boldsymbol{A}_{LL} = (\boldsymbol{L}\boldsymbol{W}^Q)(\boldsymbol{L}\boldsymbol{W}^K)^T/\sqrt{d} \qquad (13)$$

$$\boldsymbol{A}_{NN} = (\boldsymbol{N}\boldsymbol{W}^Q)(\boldsymbol{N}\boldsymbol{W}^K)^T/\sqrt{d} \qquad (14) \qquad \boldsymbol{A}_{LN} = (\boldsymbol{L}\boldsymbol{W}^Q)(\boldsymbol{N}\boldsymbol{W}^K)^T/\sqrt{d} \qquad (15)$$

Then, the value representation $\overline{\boldsymbol{L}}$ of the leaves $\boldsymbol{L}$ is computed by a linear layer, while the value representation $\overline{\boldsymbol{N}}'$ of the nodes $\boldsymbol{N}$ is encoded with tree structure using the hierarchical accumulation process (Section 4.1-4.2) as:

$$\overline{\boldsymbol{N}}' = \mathcal{V}(\mathcal{U}(\mathcal{F}(\boldsymbol{L}\boldsymbol{W}^V, \boldsymbol{N}\boldsymbol{W}^V, \mathcal{R}) + \mathsf{E}), \boldsymbol{w}) \quad ; \quad \overline{\boldsymbol{L}} = \boldsymbol{L}\boldsymbol{W}^V \qquad (16)$$

where $\boldsymbol{w} = \boldsymbol{L}\boldsymbol{u}_s$ with $\boldsymbol{u}_s \in \mathbb{R}^d$ being a trainable vector, while the weight matrices $\boldsymbol{W}^Q$, $\boldsymbol{W}^K$, $\boldsymbol{W}^V$, and $\boldsymbol{W}^O$ are similarly defined as in Section 3. After this, the resulting affinity scores for leaves and nodes are concatenated and then masked by *subtree masking* (Section 4.3) to promote bottom-up encoding. The final attention for the nodes and leaves are then computed by taking the weighted averages of the value vectors in $\overline{\boldsymbol{N}}'$ and $\overline{\boldsymbol{L}}$:

$$\mathrm{Att}_N = \mathrm{softmax}(\mu([\boldsymbol{A}_{NN}; \boldsymbol{A}_{NL}]))[\overline{\boldsymbol{N}}'; \overline{\boldsymbol{L}}] \quad (17) \quad \mathrm{Att}_L = \mathrm{softmax}(\mu([\boldsymbol{A}_{LN}; \boldsymbol{A}_{LL}]))[\overline{\boldsymbol{N}}'; \overline{\boldsymbol{L}}] \quad (18)$$

Both $\mathrm{Att}_N$ and $\mathrm{Att}_L$ are then passed through the Transformer's serial computations by function $\phi$ (Eq. 3 in Section 3), which results in the final output representations $\hat{\boldsymbol{N}}$ and $\hat{\boldsymbol{L}}$ as follows:

$$\hat{\boldsymbol{N}} = \phi(\mathrm{Att}_N \boldsymbol{W}^O, \boldsymbol{N}) \qquad (19) \qquad\qquad \hat{\boldsymbol{L}} = \phi(\mathrm{Att}_L \boldsymbol{W}^O, \boldsymbol{L}) \qquad (20)$$

**Decoder Cross-attention.** For tasks involving generation (e.g., NMT), we also use tree-based encoder-decoder attention (or cross-attention) in the decoder so that the target-side queries can leverage the hierarchical structures in the source side (tree2seq). Figure 4b shows the cross-attention process. Specifically, given the target-side query matrix $\boldsymbol{Q} \in \mathbb{R}^{t \times d}$ and the source-side leaf and node matrices $\boldsymbol{L}$ and $\boldsymbol{N}$, the affinity scores $\boldsymbol{A}_{QN} \in \mathbb{R}^{t \times m}$ and $\boldsymbol{A}_{QL} \in \mathbb{R}^{t \times n}$ are computed as:

| Model | IWSLT | | | | WMT En-De | |
|---|---|---|---|---|---|---|
| | **En-De** | **En-Fr** | **De-En** | **Fr-En** | **Base** | **Big** |
| Tree2Seq (Shi et al., 2018) | 24.01 | 40.22 | 29.95 | 39.41 | | |
| Conv-Seq2Seq (Gehring et al., 2017) | 24.76 | 39.51 | 30.32 | 39.56 | – | 25.16 |
| Transformer (Vaswani et al., 2017) | 28.35 | 43.75 | 34.42 | 42.84 | 27.30 | 29.30 |
| Dynamic Conv (Wu et al., 2019) | 28.43 | 43.72 | 34.72 | 43.08 | 27.48 | 29.70 |
| Ours | **29.47** | **45.53** | **35.96** | **44.34** | **28.40** | **29.95** |

Table 1: BLEU scores for the **base** models on IWSLT'14 English↔German, IWSLT'13 English↔French, and the **base** and **big** models on WMT'14 English→German task. Refer to Table 5 in the Appendix for parameter comparisons.

$$\boldsymbol{A}_{QN} = (\boldsymbol{Q}^t \boldsymbol{W}^Q)(\boldsymbol{N}\boldsymbol{W}^K)^T/\sqrt{d} \qquad (21) \qquad \boldsymbol{A}_{QL} = (\boldsymbol{Q}^t \boldsymbol{W}^Q)(\boldsymbol{L}\boldsymbol{W}^K)^T/\sqrt{d} \qquad (22)$$

Similar to encoder self-attention, the node representations $\boldsymbol{N}$ are encoded with the tree structure and the attention output $\text{Att}_Q$ of decoder cross-attention is computed as:

$$\overline{\boldsymbol{N}}' = \mathcal{V}(\mathcal{U}(\mathcal{F}(\boldsymbol{L}\boldsymbol{W}^V, \boldsymbol{N}\boldsymbol{W}^V, \mathcal{R}) + \mathbf{E}), \boldsymbol{w}) \; ; \; \overline{\boldsymbol{L}} = \boldsymbol{L}\boldsymbol{W}^V; \qquad (23)$$

$$\text{Att}_Q = \text{softmax}([\boldsymbol{A}_{QN}; \boldsymbol{A}_{QL}])[\overline{\boldsymbol{N}}'; \overline{\boldsymbol{L}}] \qquad (24)$$

where $\boldsymbol{w} = \boldsymbol{L}\boldsymbol{u}_c$ with $\boldsymbol{u}_c \in \mathbb{R}^d$. Note that cross-attention does not adopt subtree masking because the queries are from another domain and are not elements of the source tree.

**Remark on Speed.** Our model runs competitively with the Transformer, thanks to its *constant* parallel time complexity. In terms of *sequential* (single-CPU) computations, the hierarchical accumulation process takes $\mathcal{O}(N \log(N))$ time and our entire model maintains a time complexity identical to the Transformer, which is $\mathcal{O}(N^2)$; see Appendix 7.2 for a proof.

## 5 EXPERIMENTS

We conduct our experiments on two types of tasks: Machine Translation and Text Classification.

### 5.1 NEURAL MACHINE TRANSLATION

**Setup.** We experiment with five translation tasks: IWSLT'14 English-German (En-De), German-English (De-En), IWSLT'13 English-French (En-Fr), French-English (Fr-En), and WMT'14 English-German. We replicate most of the training settings from Ott et al. (2018) for our models, to enable a fair comparison with the Transformer-based methods (Vaswani et al., 2017; Wu et al., 2019). For IWSLT experiments, we trained the base models with $d = 512$ for 60K updates with a batch size of 4K tokens. For WMT, we used 200K updates and 32K tokens for the base models ($d = 512$), and 20K updates and 512K tokens for the big models with $d = 1024$. We parsed the texts with the Stanford CoreNLP parser (Manning et al., 2014). We used Byte Pair Encoding (Sennrich et al., 2016), where subwords of a word form a subtree. More details are provided in Appendix 7.4.

**Results.** Table 1 shows the BLEU scores for the translation tasks. Our models outperform the baselines consistently in all the tested tasks. The results demonstrate the impact of using parse trees as a prior and the effectiveness of our methods in incorporating such a structural prior. Specifically, our model surpasses the Transformer by more than 1 BLEU for all IWSLT tasks. Our big model also outdoes dynamic convolution (Wu et al., 2019) by 0.25 BLEU.

### 5.2 TEXT CLASSIFICATION

**Setup.** We also compare our attention-based tree encoding method with Tree-LSTM (Tai et al., 2015) and other sequence-based baselines on the Stanford Sentiment Analysis (SST) (Socher et al.,

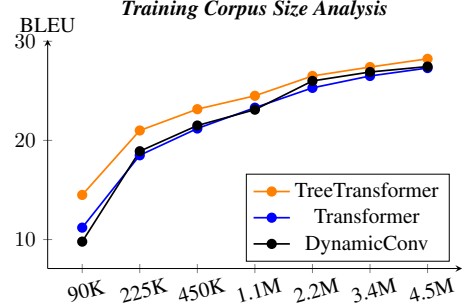

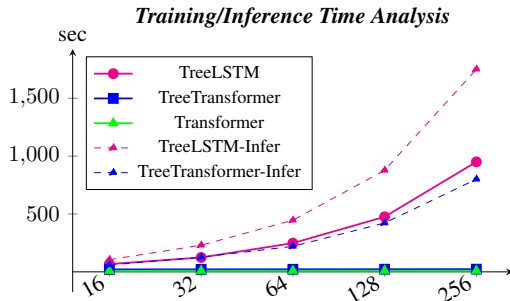

(a) WMT'14 English-German BLEU on newstest2014 with varying size of training data.

(b) Elapse training and inference time in seconds (y-axis) w.r.t sequence length (x-axis).

Figure 5: Training data size and training/inference time analysis.

| Model | En-De | En-Fr | De-En | Fr-En |
|---|---|---|---|---|
| Leaves/Nodes | 59.2/40.8 | 59.3/40.7 | 66.4/33.6 | 64.7/35.3 |
| Target→Nodes | 66.4±2e−4 | 61.9±6e−4 | 64.9±4e−4 | 59.3±2e−2 |

Table 4: Attention distributions (%) between phrases (nodes) and tokens (leaves) across different translation tasks. Statistics are derived from IWSLT'14 En-De and IWSLT'13 En-Fr test sets.

2013), IMDB Sentiment Analysis and Subject-Verb Agreement (SVA) (Linzen et al., 2016) tasks. We adopt a *tiny-sized* version of our tree-based models and the Transformer baseline. The models have 2 Transformer layers, 4 heads in each layer, and dimensions $d = 64$. We trained the models for 15K updates, with a batch size of 2K tokens. Word embeddings are randomly initialized. We provide further details of the setup in Appendix 7.4. For the Stanford Sentiment Analysis task (SST), we tested on binary (SST-2) and fine-grained (SST-5) subtasks, following Tai et al. (2015).

**Results.** Table 2 shows the results in accuracy on the classification tasks. Our Tree Transformer outperforms sequence-based Transformer and BiLSTM baselines in all tasks by a wide margin. This suggests that for small datasets, our models with a more appropriate structural bias can provide outstanding improvements compared to the vanilla Transformer. Furthermore, our models also surpass Tree-LSTM significantly in all the tested tasks, which also demonstrates our method's effectiveness compared to the best existing tree-encoding method.

| Task | Transformer | BiLSTM | Tree-Based Models | |
|---|---|---|---|---|
| | | | Tree-LSTM | Ours |
| SST-5 | 37.6 | 35.1 | 43.9 | **47.4** |
| SST-2 | 74.8 | 76.0 | 82.0 | **84.3** |
| IMDB | 86.5 | 85.8 | – | **90.1** |
| SVA | 94.4 | 95.1 | 96.2 | **98.0** |

Table 2: Classification results in accuracy (%) on Stanford Sentiment Analysis fine-grained (SST-5) and binary (SST-2), IMDB sentiment analysis, and Subject-Verb Agreement (SVA) tasks.

| Model | En-De | En-Fr | SST-5 |
|---|---|---|---|
| TreeTransformer | 29.47 | 45.53 | 47.4 |
| –HierEmb | 29.20 | 44.80 | 46.1 |
| –SubMask | 29.05 | 45.07 | 45.7 |
| –HierEmb –SubMask | 28.98 | 44.50 | 45.0 |

Table 3: Performances of different model variants on IWSLT'14 En-De, IWSLT'13 En-Fr and Stanford Sentiment Analysis (fine-grained) tasks. '–HierEmb': no hierarchical embeddings, '–SubMask': no subtree masking.

## 5.3 ANALYSIS

**Model Variations.** In Table 3, we examine the contributions of each component of our Tree Transformer on IWSLT'14 English-German translation task and Stanford Sentiment Analysis (fine-grained) task. We see that removing either or both of hierarchical embeddings and subtree masking methods has a negative impact on the model performance.

**Effectiveness on Small Datasets.** Figure 5a shows how our model performs compared to the baselines on WMT'14 English-German translation task with varying amounts of training data. It is apparent that our model yields substantial improvements (3.3 to 1.6 BLEU) when the training data is less than 1 million pairs ($< 20\%$ of WMT'14). The margin of gains gradually decreases (1.0 to 1.1 BLEU) with increasing training data. We observe similar trend in the classification tasks (Table 2), where our model outperforms sequence-based methods by around 10% absolute in accuracy. This suggests that utilizing a hierarchical architectural bias can compensate for the shortage of labeled data in low-resource scenarios.

**Training Time Analysis.** Figure 5b shows the empirical training time and inference time for the Transformer, Tree-LSTM, and our Tree Transformer with respect to input sequence length. All the models are trained on a sentiment classification task on a single GPU for 1000 iterations with a batch-size of 1. We can see that the training time for Tree-LSTM grows linearly with the sequence length. The training time for the vanilla and Tree Transformer are much less than that of the Tree-LSTM and remain relatively at a plateau with respect to the sequence length. This demonstrates our model's speed efficiency compared to Tree-LSTM or other recurrent/recursive methods. For inference, we take into account the **parsing time** of the Stanford parser ($\mathcal{O}(N)$), which substantially overwhelms the overall timing.

**Phrase- vs. Token-level Attentions.** Table 4 shows how frequently a target language token attends to phrases (nodes) vs. tokens (leaves) in the source tree. We see that although 60-66% of the source tree constituents are leaves, attentions over nodes overwhelm those over leaves (around 59% to 66%) consistently across all translation tasks, meaning that the model slightly favors phrasal attentions. The results also suggest that the attention concentrations are not correlated with the leaf/node ratio; rather, they depend on the language pairs. Leaf/node ratios might be a trivial explanation for the phenomenon, but the results indicate that certain language-dependent factors may be at play.

## 6    CONCLUSION

We presented a novel approach to incorporate constituency parse trees as an architectural bias to the attention mechanism of the Transformer network. Our method encodes trees in a bottom-up manner with constant parallel time complexity. We have shown the effectiveness of our approach on various NLP tasks involving machine translation and text classification. On machine translation, our model yields significant improvements on IWSLT and WMT translation tasks. On text classification, it also shows improvements on Stanford and IMDB sentiment analysis and subject-verb agreement tasks.

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

# 7 APPENDIX

## 7.1 PROOF FOR PROPOSITION 1

In this section, we present a proof of Proposition 1. The main argument of this proposition is that in order for $\mathcal{H}$ to be a legitimate transformation of the parse tree $\mathcal{G}(X)$, it must preserve the uniqueness of the hierarchical structure encoded by the tree. Otherwise speaking, the result of such transformation must reflect the tree $\mathcal{G}(X)$ and only $\mathcal{G}(X)$, and not any other tree form. If $\mathcal{H}$ does not have this property, it may transform two different tree structures into an identical representation, which will make the downstream tree-encoding process ambiguous. This property also means that in a parse tree (1) every node (either leaf or nonterminal node) has at most *one* parent, and (2) a nonterminal node which has multiple children can tell the sibling order between its children.

For requirement (1) specifically, except for the root node which has no parent, all other nodes have one and only one parent. Therefore, Proposition 1 implies that if there exists a inverse-transformation $\mathcal{I}$, it must be able to find the *exact* parent given any node in the tree. Our proposed transformation $\mathcal{H}$ satisfies this property. Formally, let $x \in \mathcal{L} \cup \mathcal{N}$ be a node in the tree and $\rho(x) \in \mathcal{N}$ be the parent of $x$. We define $\mathcal{P}(x) = \{y \in \mathcal{N} | x \in \mathcal{R}(y)\}$ to be the set of all ancestors of $x$. Thus, the parent of $x$ should belong to $\mathcal{P}(x)$. From that, we derive the parent $\rho(x)$ of $x$ as:

$$\rho(x) = y_x \in \mathcal{P}(x) \text{ such that } \mathcal{R}(y_x) \cap \mathcal{P}(x) = \{y_x\} \tag{25}$$

As implied in Equation 25, the parent $\rho(x)$ is one of the ancestors of $x$ whose subtree does not encapsulate any of the ancestors in $\mathcal{P}(x)$ except itself. In other words, the parent is the immediate ancestor. In a parse tree, there is *at most one* node $y_x$ that satisfies this condition. We prove this by contradiction below.

**Proof 1** *Suppose there exists $\rho'(x) = y'_x \neq y_x$ such that $\mathcal{R}(y'_x) \cap \mathcal{P}(x) = \{y'_x\}$, i.e., $x$ has 2 parents. Under this assumption, there will be two different paths from the root node to $x$ – one via $y_x$ and the other via $y'_x$. This makes a closed cycle. However, a parse tree is an acyclic directed spanning tree, which does not have cycles. Thus, by contradiction, if $\mathcal{G}(x)$ is a spanning tree reflecting the true parse tree, then $y'_x = y_x$.*

For requirement (2), we can prove it by exploiting the fact that in the definition of $\mathcal{T}(X) \overset{\Delta}{=} (\mathcal{L}, \mathcal{N}, \mathcal{R})$, $\mathcal{L}$ is *ordered* according to the original word order in the text. Generally speaking, if a nonterminal node $x$ has $t$ children $(c^1, ..., c^t)$, each child node $c^k$ heads (or bears) its own subtree $\mathcal{R}(c^k)$ which includes certain leaves $\mathcal{R}(c^k) \cap \mathcal{L}$. Then, the order of leaves $\mathcal{L}$ indicates the order of different leaf sets $\mathcal{R}(c^k) \cap \mathcal{L}$, which in turn indicates the sibling order of the children of node $x$.

Formally, let $\mathcal{L} = (l_1, ..., l_n)$ where $l_i$ is the $i$-th word in the input text, and $x$ is an arbitrary nonterminal node whose children are $(c^1, ..., c^t)$ (for $t > 1$) with any two of them being $c^k$ and $c^{k'}$ ($c^{k k'}$), then either:

$$\begin{cases} (i < j \ \forall l_i \in \mathcal{R}(c^k) \cap \mathcal{L} \text{ and } \forall l_j \in \mathcal{R}(c^{k'}) \cap \mathcal{L}) \text{ or} \\ (i > j \ \forall l_i \in \mathcal{R}(c^k) \cap \mathcal{L} \text{ and } \forall l_j \in \mathcal{R}(c^{k'}) \cap \mathcal{L}) \end{cases} \tag{26}$$

We proceed to prove the first clause in Equation 26, while the second clause can be similarly inferred.

Specifically, the first clause in Equation 26 implies that if there exists a leaf $l_{i'} \in \mathcal{R}(c^k) \cap \mathcal{L}$ and a leaf $l_{j'} \in \mathcal{R}(c^{k'}) \cap \mathcal{L}$ such that $i' < j'$, then $i < j \ \forall l_i \in \mathcal{R}(c^k) \cap \mathcal{L}$ and $\forall l_j \in \mathcal{R}(c^{k'}) \cap \mathcal{L}$, or in other words, subtree $\mathcal{R}(c^k)$ is to the left side of subtree $\mathcal{R}(c^{k'})$. We prove this by contradiction:

**Proof 2** *Suppose there exist $l_{i'}, l_{i*} \in \mathcal{R}(c^k) \cap \mathcal{L}$ and $l_{j'}, l_{j*} \in \mathcal{R}(c^{k'}) \cap \mathcal{L}$ such that $i' < j'$ but $i* \geq j*$ and not both $i' = i*$ and $j' = j*$. If $i* = j*$, this creates a closed cycle, which is impossible as described in proof 1. If $i* > j*$, $l_{j*}$ lies between the leftmost and rightmost leaves of $\mathcal{R}(c^k) \cap \mathcal{L}$. As a result, the subtree $\mathcal{R}(c^k)$ does not cover a full contiguous phrase but multiple segmented phrases, which is prohibited for a parse tree. Therefore, by contradiction, there can not exist $l_{i*} \in \mathcal{R}(c^k) \cap \mathcal{L}$ and $l_{j*} \in \mathcal{R}(c^{k'}) \cap \mathcal{L}$ such that $i* \geq j*$.*

In general, combing the satisfactions from the above requirements, the reverse-transformation $\mathcal{I}$ can convert $\mathcal{T}(X) = (\mathcal{L}, \mathcal{N}, \mathcal{R})$ back to the original graph $\mathcal{G}(X)$:

$$\mathcal{I}(\mathcal{H}(\mathcal{G}(X))) = \mathcal{G}(X) \tag{27}$$

## 7.2 TIME COMPLEXITY ANALYSIS

In this section, we prove that given single-thread sequential computation (single CPU), our method operates at $\mathcal{O}(N^2)$ time complexity. In short, the hierarchical accumulation process (Section 4.1) runs at $O(N \log N)$ time, while the attention scores ($\boldsymbol{QK}^T$) in standard attention are computed at $\mathcal{O}(N^2)$. Overall, the tree-based attention can performs at $\mathcal{O}(N^2)$ time complexity, same as the Transformer. We then proceed to analyze the time complexity of the hierarchical accumulation process. Formally, let $X$ be a $n$-length sentence with $\mathcal{T}(X) = (\mathcal{L}, \mathcal{N}, \mathcal{R})$ as its balance binary constituency parse tree as defined in section 4, $\boldsymbol{N} \in \mathbb{R}^{m \times d}$ and $\boldsymbol{L} \in \mathbb{R}^{n \times d}$ be the hidden states of elements in $N$ and $L$ respectively. Therefore, there are $m = n - 1$ non-terminal nodes in the tree, which is also the size of $\mathcal{N}$. In the *upward cumulative-average* operation $\mathcal{U}$, each branch from the root to a leaf has $\approx \log(n)$ nodes, the cumulative operation of these nodes can be done at $\mathcal{O}(\log(n))$. That is, because the result $y_i$ of each node $x_i$ can be computed as $y_i = y_{i-1} + x_i$, computations for all nodes in a branch take linear time using dynamic programming, yielding $\mathcal{O}(\log(n))$ time complexity. As we have $n$ branches in the tree, the total complexity for $\mathcal{U}$ is $\mathcal{O}(n \log(n))$. Likewise, the *weighted aggregation* operation $\mathcal{V}$ is also computed at $\mathcal{O}(n \log(n))$ complexity. Specifically, at level $i$ from the root of the tree, there $i$ non-terminal nodes, which each has to aggregate $n/i$ components $\hat{\mathbf{S}}_{i,j}$ to calculate the final representation of the node. Thus, at each level, there are $n$ computations. Because the total height of the tree is $\log(n)$, the time complexity of $\mathcal{V}$ is also $\mathcal{O}(n \log(n))$. Hence, the total complexity of hierarchical accumulation process is $\mathcal{O}(n \log(n))$. As such, the final *sequential* time complexity of the proposed attention layer is:

$$\mathcal{O}(N^2) + \mathcal{O}((N-1)^2) + \mathcal{O}((N-1)N) + \mathcal{O}(N \log(N)) = \mathcal{O}(N^2) \tag{28}$$

Having said that, in the era that powerful GPU-based hardware is ubiquitous, it is important to note that our models can achieve comparable parallelizability compared to the Transformer, while they can leverage the essence of hierarchical structures in natural languages. The purpose of the above time complexity analysis is only to show more insights about our models. That being said, even though we provide numerical analysis of training speed (figure 5b) as objective as we could, training speed depends greatly on different subjective factors that may stray the actual timing away from its theoretical asymptotic time complexity. These factors are data preprocessing and batching, actual low-level and high-level programmatic implementation of the method, GPUs, CPUs, I/O hardwares, etc. For instance, a standard LSTM module in Tensorflow or Pytorch may performs 10 times slower than an LSTM with CUDA CuDNN kernels, even though no extra computations are theoretical required.

## 7.3 OVERALL ARCHITECTURE

Figure 6 shows the overall Seq2Seq architecture of our model. In the encoder specifically, we parse the source sequence into constituency trees and then feed the leaf and node components into a stack of encoder layers. The leaf and node components are passed through the tree-based self-attention layer, where the value representation of node is incorporated with hierarchical accumulation. After that, the output representations of leaves and nodes are passed through a weight-shared series of layer normalizations and and feed-forward layer. In the decoder, the query representation of the target domain attends on the leaves and nodes representation of the source sequence as computed by the encoder. For more insights, apply hierarchical accumulation on the key components (instead on value components) causes dramatic performance because they are disrupted after being multiplied with the query components and they do not directly contribute to the representations of the outputs. Meanwhile, applying accumulation on attention scores does not yield performance benefits but increases computational burden.

Table 5 shows the exact number of parameters of the Transformer and our method. As it can be seen, the increase of parameters is almost unnoticeable because the hierarchical embedding modules share weights among different attention heads.

## 7.4 MORE DETAILED TRAINING CONFIGURATIONS

**Text Classification.** We adopt the *tiny* size versions of our tree-based models as well as the transformer baseline. The models possess 2 transformer-layers, each has model dimension of 64, 4 heads. We trained the models for $15,000$ updates, with batch size of 2048 tokens. We

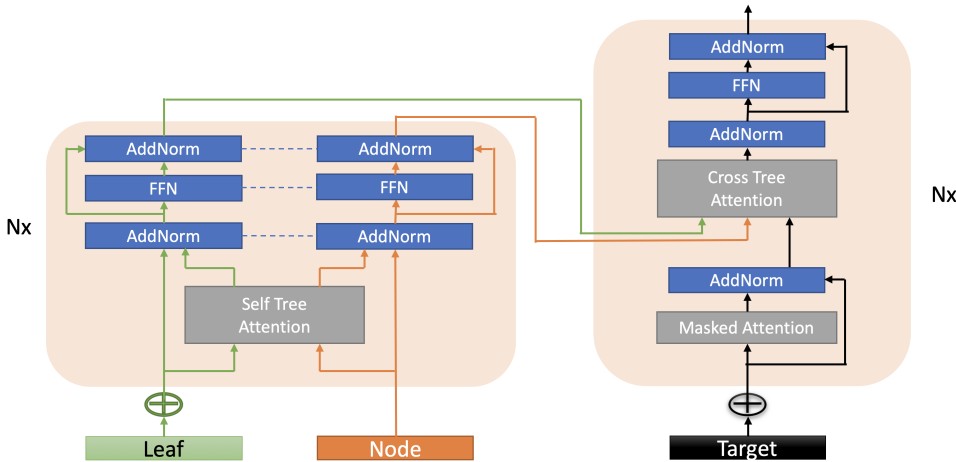

Figure 6: Overall architecture of Tree Transformer. (Dashed lines: sharing parameters)

| Model | Base | Big |
|---|---|---|
| Transformer | 61,747,200 | 209,813,504 |
| Ours | 61,810,944 (+0.1%) | 209,967,104(+0.07%) |

Table 5: Exact number of parameters for Transformer and our model, both used for WMT'14 English-German task.

used random initialized embeddings for all experiments. For the Stanford Sentiment Analysis task (SST), we tested on two subtasks: binary and fine-grained (5 classes) classification on the standard train/dev/test splits of 6920/872/1821 and 8544/1101/2210 respectively. We optimize every sentiment label provided in the dataset. We used a learning rate of $7 \times 10^{-4}$, dropout 0.5 and 8000 warmup steps. For subject-verb agreement task, we trained on a set of $142,000$ sentences, validated on the set of $\approx 15,700$ sentences and tested on the set of $\approx 1,000,000$ sentences. For IMDB sentiment analysis , we used a training set of $25,000$ documents and test set of $25,000$ documents. As the documents are multi-sentence, they are added with a dummy root node which is used to predict the sentiment. For both subject-verb agreement and IMDB sentiment analysis, we trained models with 20 warmup steps, 0.01 learning rate and 0.2 dropout.

**Machine Translation.** We replicate most of the *base* training settings from Ott et al. (2018) for our models, to enable a fair comparison with transformer-based methods (Vaswani et al., 2017; Wu et al., 2019). For IWSLT experiments, we trained the models with $d = 512$, feed-forward dimension $d_{ffn} = 1024$, approximate batch size of $4000$ tokens, $60,000$ updates, learning rate $5 \times 10^{-4}$, 4000 warmup steps, dropout rate 0.3, L2 weight decay $10^{-4}$, beam size 5 and length penalty 1.0 for English-German and 0.7 for English-French. The hierarchical embedding size $|E|$ is set to 100. We used BPE tokens pre-trained with $32,000$ iterations. Note that if a word is broken into multiple BPE subwords, it form a subtree with leaves as such subwords and root-node as the POS tag of the word. Figure 7 visualizes how this works. The IWSLT English-German training dataset contains $\approx 160,000$ sentence pairs, we used $5\%$ of the data for validation and combined (IWSLT14.TED.dev2010,dev2012,tst2010-tst2012) for testing. Meanwhile, the IWSLT English-French task has $\approx 200,000$ training sentence pairs, we used IWSLT15.TED.tst2012 for validation and IWSLT15.TED.tst2013 for testing. We use the Stanford CoreNLP parser (v3.9.1)[5] (Manning et al., 2014) to parse the datasets. For WMT experiments, we trained the models with $d_{model} = 512$, feed-forward dimension $d_{ffn} = 2048$, batch size of $\approx 32,000$ tokens, $200,000$ updates, learning rate $7 \times 10^{-4}$, 4000 warmup steps, dropout rate 0.1, L2 weight decay $10^{-4}$, beam size 5 and length penalty 0.6. We take average of the last 5 checkpoints for evaluation. The WMT"14 English-German dataset contains $\approx 4.5M$ pairs. we tested the models on newstest2014 test set. We used tokenized-BLEU to evaluate the models.

---

[5]http://nlp.stanford.edu/software/stanford-corenlp-full-2018-02-27.zip

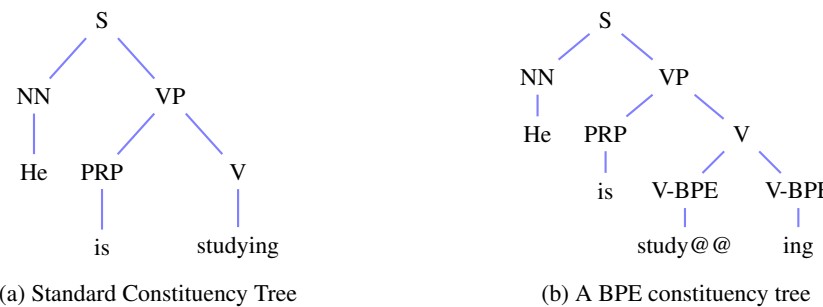

(a) Standard Constituency Tree        (b) A BPE constituency tree

Figure 7: Process to break standard tree (fig. 7a) into BPE tree (fig. 7b).

**Training Time Analysis.** For this experiment, all the examined models (Transformer, Tree Transformer and Tree-LSTM) are trained on text classification task on a GPU for $1000$ training steps with batch-size of $1$. For vanilla and Tree Transformer, we used the 2-layer encoder with dimension $64$. For Tree-LSTM, we used one-layer LSTM with dimension 64. A binary constituency tree is built given each input sentence. The parser's processing time is significant. For training time, we exclude the parser's time because the parser processes the data only once and this counts towards to preprocessing procedures of the dataset. However, we include parser's time into inference time because we receive surface-form text instead of trees. In practice, the parsing time substantially overshadows the computation time of the network. Thus, the overall process will be faster if a more efficient parser is developed.

**Effectiveness on Small Datasets.** We used both Transformer and Tree Transformer on *base*-size training settings from Ott et al. (2018). We trained the models with batch size of $32,000$ tokens, $100,000$ updates, learning rate $7 \times 10^{-4}$, $4000$ warmup steps, dropout $0.1$, L2 weight decay $10^{-4}$, beam size $5$ and length penalty $0.6$. For both sets of tasks, we take average of the last 5 checkpoints for evaluation. We used BPE tokens pre-trained with $32,000$ iterations. We randomly samples the training data into size-varying portions: 2%(90K pairs), 5%(225K pairs), 10%(450K pairs), 25%(1.125M pairs), 50%(2.25M pairs), 75%(3.375M pairs) and 100%(4.5M pairs).

