# OpenReview forum: "Tree-Structured Attention with Hierarchical Accumulation"
_ICLR.cc/2020/Conference — Accept (Poster)_

### Official Review · AnonReviewer2 · 2019-10-22
**Official Blind Review #2**

**Rating:** 8

**Review:**

This paper extends transformers by enabling them to incorporate hierarchical structures like constituency trees structured attention with hierarchical accumulation. In particular, they modified the architecture of transformers such that they can learn phrase-level attention scores, and use them in the final assigned task of text classification or language translation.
They also explored a couple of variants of their proposed architecture: Hierarchical embeddings and Subnet masking which helped them outperform SOTA methods including Tree-LSTM (similar to this paper in principle, except that it was designed for LSTM-based models and not transformers).

The paper is well written and well-augmented with supportive figures. Particularly, Figure1 was very helpful in understanding all the complexities of the proposed model. Further, the authors justify the utility of the proposed approach covering different aspects of evaluation including comparative studies with baselines, ablation studies, phrase vs token-level attentions, training-time analysis.

A limitation of this work is high inference cost. As the results indicate, parsing trees from text is the most costly step in the entire framework, and consequently, the inference time of proposed model will still be much higher than transformers. Hence, this work might still not be applicable to low-latency constraint scenarios.


Other Comments:
1) I did not fully understand why it would be better to mask out the non descendants in subnet masking approach. Why shouldn't a phrase node seek attention from tokens outside its scope? Probably the answer lies in the the way these trees are constructed. Nevertheless, it would be useful to provide some intuition with examples to motivate subtree masking.

2) In Equation(5), the subscripts "i-1" should be replaced with "i"? Otherwise it will be sensitive to ordering of non-terminal nodes in N, and also Figure 1 wouldn't make sense.


**Experience Assessment:**

I have read many papers in this area.

**Review Assessment: Checking Correctness Of Derivations And Theory:**

I carefully checked the derivations and theory.

**Review Assessment: Checking Correctness Of Experiments:**

I assessed the sensibility of the experiments.

**Review Assessment: Thoroughness In Paper Reading:**

I read the paper thoroughly.

---

### Official Review · AnonReviewer1 · 2019-10-23
**Official Blind Review #1**

**Rating:** 6

**Review:**

Summary: This paper describes an attention-based method to encode trees with constant parallel-time complexity maintaining scalability, uses this model to encode constituency parses of input sentences for the tasks of machine translation and text classification, and shows improved accuracy/bleu over models that do not encode the parses.

Strengths:

The method proposed by the authors is scalable despite encoding tree structures. The models give considerable improvements on various machine translation datasets and the authors also show that the model is more sample efficient. I appreciate the charts showing training and inference time with sentence length, and the tables showing the ablations and attention distributions.

Weaknesses

The authors do not use pre-trained embeddings for any of the classification models, but using these embeddings boost performance to much more than what they authors have achieved here. My main question is, if we use pre-trained embeddings, do encoding constituency parses add anything over and above them? I would like to see this method improve results over some state of the art classification models and not just over the tree-LSTM. In other words, how much classification performance does this method yield over current SOTA models, because the SST results are quite a bit below the current SOTA.

The authors achieve an accuracy of 98.2 on the IMDB dataset. Is this actually the case or is this a bug? Even models using BERT barely achieve an accuracy of 96% (http://nlpprogress.com/english/sentiment_analysis.html). Or am I looking at the wrong dataset here? Can the authors clarify?


**Experience Assessment:**

I have published in this field for several years.

**Review Assessment: Checking Correctness Of Derivations And Theory:**

I did not assess the derivations or theory.

**Review Assessment: Checking Correctness Of Experiments:**

I carefully checked the experiments.

**Review Assessment: Thoroughness In Paper Reading:**

I read the paper at least twice and used my best judgement in assessing the paper.

---

### Official Review · AnonReviewer3 · 2019-10-24
**Official Blind Review #3**

**Rating:** 6

**Review:**

This paper proposes a novel mechanism to leverage additional tree-structure information into the transformer. The proposed method consist three main component: a hierarchical accumulation strategy to aggregate the leaf node embedding information to the non-terminal nodes; a hierarchical embedding which is akin to positional embedding, however encapsulating the leaf-to-node and relative position information; a sub-tree masking strategy to filter out irrelevant information. Although the formulations seem a little bit overcomplicated/cumbersome to me, I find the figures are really helpful to understand the gists, which are intuitive and straightforward.

Strengths:

* Intuitive and novel ideas. The tree structure information is incorporated into the transformer in a sophisticated and elegant manner, with only constant time-complexity overhead.

* Strong empirical results and well-defined ablation study. Good analysis on the performance vs dataset size and time complexity

Cons:

* Heavy notations which could be more concise.

* Implementing such an architecture without source code could be difficult.

Questions:

* Comparing with the naive transformer architecture (e.g. base model), how many additional parameters are there in your model? I see that for fair comparison, the authors use the same base transformer architecture, however it would still be very helpful if they can provide the statistics of the number of parameters for the proposed model and the compared baselines.

* Are all the baselines using the same parser (CoreNLP)? If not, would the difference of parsing trees be a confounding factor?





**Experience Assessment:**

I have read many papers in this area.

**Review Assessment: Checking Correctness Of Derivations And Theory:**

I carefully checked the derivations and theory.

**Review Assessment: Checking Correctness Of Experiments:**

I assessed the sensibility of the experiments.

**Review Assessment: Thoroughness In Paper Reading:**

I read the paper thoroughly.

---

### Public Comment · ~Zihao_Ye1 · 2019-12-28
**About the result on text-classification**

As far as I know, datasets like IMDB did not provide golden constituency parsing trees of each sentence, so you might use some parser(such as corenlp, etc) to get the parse trees, and there would be many errors. How do Transformers benefit from an error-prone parse tree so much (10% on IMDB)? Some fine-grained analysis or ablation studies would make the result more convincing.

What confuses me most is that your result on SST is too low (47.4 on test acc, as a reference, Tree-LSTM can get a 51.0(±0.7) with the help of pre-trained GloVe embeddings and you report 43.9), if it's because you use random-initialized word embeddings, then why your result on IMDB is extremely high (98.2)? as reviewer 1 mentioned, even BERT can only achieve an accuracy of 96.

I've experimented with transformer on tree-like hierarchical structures (https://rlgm.github.io/papers/67.pdf , ICLR-RLGM 2019) before, our model also updates hierarhical features with transformers, the difference is: 1. use naive binary tree or constituency parse tree 2. sparse or dense attention 3. w/ or w/o hierarchical aggregation.
Considering the difference, I changed my structure to use constituency parse tree(produced by corenlp) and make the  attention dense, however I did not observe performance gain on IMDB.

I would appreciate it if authors could provide more ablation studies (e.g. what if you use naive binary tree instead of constituency parse trees with your structure, the effect of hierarchical aggregation), thanks.

---

### Decision · Program_Chairs · 2019-12-19

**Decision:**

Accept (Poster)

**Comment:**

This paper incorporates tree-structured information about a sentence into how transformers process it. Results are improved. The paper is clear. Reviewers liked it. Clear accept.